# High Body Mass Index Disrupts the Homeostatic Relationship Between Pain Inhibitory Control and the Symptomatology in Patients with Knee Osteoarthritis—A Cross-Sectional Analysis from the DEFINE Study

**DOI:** 10.3390/neurosci6010014

**Published:** 2025-02-08

**Authors:** Guilherme J. M. Lacerda, Felipe Fregni, Linamara R. Battistella, Marta Imamura

**Affiliations:** 1Neuromodulation Center and Center for Clinical Research Learning, Spaulding Rehabilitation Hospital and Massachusetts General Hospital, Harvard Medical School, Boston, MA 02138, USA; 2Instituto de Medicina Física e Reabilitação, IMREA, Hospital das Clínicas HCFMUSP, Faculdade de Medicina FMUSP, Universidade de São Paulo, São Paulo 04116-040, Brazil; linamara@usp.br (L.R.B.);; 3Departamento de Medicina Legal, Bioética, Medicina do Trabalho e Medicina Física e Reabilitação, Faculdade de Medicina da Universidade de São Paulo (FMUSP), São Paulo 01246-903, Brazil

**Keywords:** knee osteoarthritis, body mass index, BMI, quantitative sensory testing, pain, CPM, conditioned pain modulation

## Abstract

Objective: As outlined in our previous study, this study aims to investigate the role of body mass index (BMI) as an effect modifier in the relationship between conditioned pain modulation (CPM) and clinical outcomes, including depression, quality of life, and pain in individuals with knee osteoarthritis (KOA). Methods: This cross-sectional analysis is part of the DEFINE Study in Rehabilitation. A total of 113 participants with KOA, admitted to the Instituto de Medicina Física e Reabilitação (IMREA) rehabilitation program, were included. Clinical and neurophysiological assessments were conducted, focusing on CPM, the Hamilton Depression Rating Scale (HDRS), and the SF-36 health survey. BMI was stratified into two categories based on the mean BMI of 31.99 kg/m^2^, and linear regression models were used to evaluate BMI as an effect modifier in the relationship between CPM and clinical outcomes. *p*-values below 0.10 for interaction terms (CPM × BMI) indicated effect modification. Results: In participants with BMI < 31.99 kg/m^2^, increased CPM was significantly associated with improved depression scores (lower HDRS) and enhanced physical functioning, emotional well-being, and reduced limitations due to emotional problems (SF-36). In contrast, no significant associations between CPM and these outcomes were found in participants with BMI ≥ 31.99 kg/m^2^. The results suggest that a higher BMI disrupts the salutogenic effects of endogenous pain control, diminishing the beneficial associations between CPM and both physical and psychological outcomes, as previously observed in fibromyalgia patients. Conclusions: BMI acts as an effect modifier in the relationship between CPM and clinical outcomes in individuals with KOA. Obesity appears to hinder the beneficial relationships between clinical symptoms and CPM, leading to a less favorable link between physical and emotional functioning and CPM. These findings highlight the importance of considering BMI in treatment strategies for KOA, particularly when addressing the impact of lifestyle and other modifiable factors that influence pain modulation.

## 1. Introduction

Osteoarthritis is the most prevalent form of arthritis among adults, marked by persistent pain and reduced mobility, with its occurrence rising sharply with age. The WHO has designated 2021–2030 as the Decade of Healthy Ageing, emphasizing the importance of addressing conditions like osteoarthritis that significantly impair functional capacity and quality of life. Additionally, osteoarthritis often coexists with other chronic illnesses, exacerbating their negative effects [1]. Specifically, knee osteoarthritis (KOA) is a widespread chronic joint condition, affecting more than 654 million people globally and imposing substantial societal costs, especially among older adults [1,2]. Knee osteoarthritis (KOA), a form of osteoarthritis, is a degenerative joint disease characterized by joint pain, stiffness, and functional impairments, leading to decreased mobility and diminished quality of life. OA primarily manifests as the degeneration of joint cartilage and is most commonly observed in the knees and hips, although it can affect any joint. The condition predominantly affects individuals over 50 years of age and poses significant challenges to daily functioning and overall well-being [3]. As populations continue to age and obesity rates tend to rise concurrently, the combined effects of these trends exacerbate the burden of conditions like KOA [4,5].

Given these trends, it becomes crucial to utilize reliable assessment tools to evaluate the impact of KOA on both physical function and mental well-being. One important aspect to be considered is the effect modifiers. Effect modification is used in clinical research to determine if a treatment or exposure has different effects across patient groups with distinct characteristics. This approach helps identify subgroups at higher or lower risk, those who may benefit most, or those unlikely to benefit from a treatment, guiding more personalized interventions [6]. In this context, a meta-analysis revealed that chronic pain patients exhibit altered quantitative sensory testing (QST) compared to healthy controls, with reduced conditioned pain modulation (CPM) being a key characteristic. CPM refers to the phenomenon by which a conditioning stimulus induces a change in the perception of a test stimulus, reflecting the net effect of complex facilitatory and inhibitory mechanisms of pain processing, as standardized in human psychophysical research [7]. These findings indicate a shift in the pain processing mechanisms in individuals with chronic pain, suggesting impaired endogenous pain inhibition and altered sensitivity to pain stimuli [8]. Moreover, recent studies suggest that BMI acts as an effect modifier in the relationship between CPM and clinical outcomes such as depression, pain, and quality of life in fibromyalgia patients [9,10]. Similarly, research has demonstrated that obesity is a risk factor for pain in diabetes, with findings indicating that central obesity weakens pain inhibition [11]. This indicates that body weight may influence how effectively the pain modulation system functions and its impact on these key health outcomes.

Despite significant progress in understanding knee osteoarthritis (KOA) and its related pain mechanisms, there remains a crucial gap in knowledge about how BMI affects endogenous pain control, particularly through conditioned pain modulation (CPM). Although BMI has been recognized as a potential modifier of clinical outcomes in other conditions [12,13], the specific ways in which body weight influences pain modulation in KOA patients are still unclear. This gap hinders the development of personalized treatment strategies for patients with different BMI levels, limiting the effectiveness of rehabilitation and pain management in KOA. To address this gap for KOA patients, we followed a similar approach to our previous study on fibromyalgia patients [9], where we investigated the role of BMI in disrupting endogenous pain modulation mechanisms. Specifically, in this study, we examine the role of BMI as an effect modifier in the relationship between conditioned pain modulation (CPM) and clinical outcomes such as depression, quality of life, and pain intensity in individuals with KOA. We hypothesize that individuals with lower BMI levels exhibit more effective endogenous pain control, leading to better clinical outcomes, whereas those with higher BMI levels show reduced CPM efficiency. This study aims to explore these associations to provide groundwork for understanding the potential influence of BMI on pain modulation in KOA.

## 2. Methods

### 2.1. Participants, Study Design, and Sample Size

This research is part of the Deficit of Inhibition as a Marker of Neuroplasticity DEFINE Study in Rehabilitation, a Longitudinal Cohort Study Protocol [14]. Patients who were admitted to the DEFINE rehabilitation program at the Instituto de Medicina Física e Reabilitação (IMREA) were invited to participate. A total of 113 participants provided informed consent, which had been previously approved by the Ethics Committee for Research Protocol Analysis at the Hospital das Clínicas, Faculdade de Medicina, Universidade de São Paulo (CAAE: 86832518.7.0000.0068). The participants underwent both clinical and neurophysiological assessments at two time points: before and after the IMREA rehabilitation program. For this cross-sectional analysis, only the pre-intervention assessments were used. To analyze the data, we calculated the sample size by detecting clinically meaningful differences in pain and CPM. For pain, the minimally clinically important difference (MCID) was set to correspond to Cohen’s d = 0.5, requiring a sample size of 34 participants. For CPM, a 30% effect or Cohen’s d = 0.3 was used, requiring a sample size of 90 participants. With 113 participants, our study exceeds these requirements, ensuring sufficient power to detect clinically significant differences. These parameters align with those established in our previous research on fibromyalgia patients [9].

### 2.2. Inclusion Criteria

Participants of both sexes were included in the study if they were over 18 years of age, were clinically stable as verified by medical evaluation, signed the informed consent form, and met the eligibility criteria for the IMREA rehabilitation program. Specifically, participants in the knee OA group were required to have a clinical diagnosis of knee osteoarthritis, confirmed by radiological imaging (bilateral knee radiography).

### 2.3. Exclusion Criteria

Participants were excluded if pregnant, had active OA in joints other than the knee, or had any clinical or social conditions that interfered with participation in rehabilitation treatment.

### 2.4. Pain-Related Variables

We used quantitative sensory testing (QST) for experimental pain assessment. For the Pressure Pain Thresholds (PPTs), with the patient seated, measurements were taken three times on the right hand to quantify pain sensitivity levels, with a 15 s interval between each measurement. The algometer was applied to the muscle belly in the thenar region, toward the wrist. The patient was asked to report the first noticeable signs of pain, at which point the force was immediately stopped, and the value was recorded. The average of the three measurements was then calculated and documented. For the conditioned pain modulation (CPM), we used a protocol based on changes in PPTs [15]. Participants immersed their left hand in cold water (10–12 °C) for one minute, ensuring that the hand and wrist were fully submerged (with the water level approximately 5 cm above the wrist). The first algometry measurement was then taken on the contralateral (right) hand after 30 s of immersion, followed by a second measurement at 45 s, and a third at 60 s. The average of these three measurements was documented.

To calculate CPM, the average algometry measurement taken before immersion (on the right hand) was subtracted from the average measurement taken on the right hand during immersion. This procedure was repeated bilaterally, and the averages from both sides were taken to calculate the final CPM value for the study. This procedure was performed bilaterally, and the average of the measurements was taken to calculate the CPM.

### 2.5. Clinical Variables

We chose to cover important aspects related to knee osteoarthritis, namely depression and impact on quality of life [16,17]. For this purpose, we selected the following variables as outcomes:

### 2.6. Hamilton Depression Rating Scale (HDRS)

The Hamilton Depression Rating Scale, most used in its 17-item version in clinical research, is designed to assess a wide range of depressive symptoms. Each item is scored on a scale of 0 to 2, depending on the specific symptom. The symptoms evaluated include depressed mood, feelings of guilt, suicidal thoughts, insomnia, work and activity levels, psychomotor retardation or agitation, anxiety, somatic symptoms, weight loss, and cognitive impairment. Together, these items provide a comprehensive assessment of depression severity. A comprehensive meta-analysis, spanning studies from 1960 to 2008, offered valuable insights into the reliability of the HAM-D [18]. It demonstrated that the scale performs consistently across various reliability measures, especially in terms of inter-rater reliability, showing that it yields stable results when administered by different clinicians. Overall, the HAM-D is a well-established, reliable tool for assessing depression.

### 2.7. SF-36 Questionnaire (SF-36)

The SF-36 Questionnaire is a generic, multipurpose health survey used to assess health-related quality of life. Originally developed as part of the Medical Outcomes Study, it has been extensively validated and translated into over 50 languages. The SF-36 consists of items organized into eight domains: physical functioning, role limitations due to physical health, bodily pain, general health perceptions, vitality, social functioning, role limitations due to emotional problems, and mental health. The questionnaire generates two composite scores—physical and mental health summary measures—allowing for a comprehensive evaluation of both physical and mental aspects of health. Each domain is scored on a scale from 0 to 100, with higher scores indicating better health status [19].

### 2.8. Body Mass Index (BMI)

The body mass index (BMI) is the most widely used measure to evaluate weight relative to height and classify obesity in adults. As defined by the World Health Organization (WHO), BMI is calculated by dividing an individual’s weight in kilograms by the square of their height in meters (kg/m^2^) [20].

### 2.9. Statistical Analysis

We conducted linear regressions separately with CPM as the independent variable and HDRS and SF36 domains as dependent variables. Afterwards, we added an interaction term (CPM*BMI_mean) to evaluate BMI as a potential effect modifier. BMI_mean is a binary variable representing BMI, divided into two categories based on the average BMI of 31.99 kg/m^2^: below the mean and above the mean. BMI was evaluated as a potential effect modifier if the interaction term showed a *p*-value < 0.10. For statistically significant interaction terms, linear regression analyses were performed separately for each BMI group to examine the relationships between the dependent and independent variables. In the analysis within BMI groups, *p*-values below 0.05 were considered statistically significant. All statistical analyses were performed using R-Studio Version 2023.06.0+421.

## 3. Results

### 3.1. Sample Characteristics

Table 1 summarizes the characteristics of the sample, consisting of 113 participants. The mean age of the participants is 68.7 years, with a standard deviation (SD) of 9.45 years. The sample includes 19 males (16.8%) and 94 females (83.2%). Educational levels vary among the participants: 2 (1.8%) are illiterate, 48 (42.3%) have completed elementary school, 34 (30.1%) have completed high school, and 29 (25.7%) have a college degree or higher. In terms of ethnicity, the majority, 72 participants (63.7%), identify as White. This is followed by 22 participants (19.5%) who identify as Multiracial, 13 participants (11.5%) who identify as Black or African American, and 6 participants (5.3%) who identify as Asian. The mean duration of the disease among the participants is 95.7 months, with a standard deviation of 98.8 months. Additionally, the mean body mass index (BMI) is 31.99 kg/m^2^, with a standard deviation of 5.30 kg/m^2^.

The distribution of BMI categories among the participants shows that 57 participants have a BMI less than 31.99 kg/m^2^, 51 participants have a BMI greater than 31.99 kg/m^2^, and there are missing data for 5 participants.

### 3.2. Variable Characteristics

Table 2 outlines the variable characteristics. Conditioned pain modulation has a mean of −0.63 (95% CI: −1.03–−0.23, SD: 2.01). The Hamilton Depression Rating Scale has a mean of 9.36 (95% CI: 8.31–10.41, SD: 5.58). The SF-36—Physical Functioning has a mean of 40.33 (95% CI: 36.09–44.57, SD: 22.37). The SF-36—Role Limitations due to Emotional Problems has a mean of 51.71 (95% CI: 43.07–60.35, SD: 45.61). The SF-36—Emotional Well-being has a mean of 68.5 (95% CI: 64.57–72.43, SD: 20.74). The SF-36—Pain has a mean of 39.35 (95% CI: 34.95–43.75, SD: 23.23).

### 3.3. BMI as an Effect Modifier

Table 3 summarizes the statistically significant *p*-values for the interaction term of CPM and BMI_mean, affecting the dependent variables: HDRS, SF-36 (Physical Functioning), SF-36 (Role Limitations due to Emotional Problems), SF-36 (Emotional Well-being), and SF-36 (Pain).

### 3.4. BMI as an Effect Modifier in the Relationship Between CPM and Depression

Using a BMI cutoff of 31.99 kg/m^2^, we stratified the sample into two groups: BMI < 31.99 and BMI ≥ 31.99 kg/m^2^. In the group with a BMI < 31.99, a statistically significant negative association was found between CPM and HDRS (β-coefficient = −0.98; *p*-value = 0.017), suggesting that an increase in CPM corresponds to a decrease in depression scores. However, this relationship was not observed in the BMI ≥ 31.99 group, where no statistical significance was detected (*p*-value = 0.93).

### 3.5. BMI as an Effect Modifier in the Relationship Between CPM and SF-36

Using a BMI mean cutoff of 31.99 kg/m^2^, we stratified the sample into two groups: BMI < 31.99 and BMI ≥ 31.99 kg/m^2^. In the group with a BMI < 31.99, significant associations were observed between CPM and multiple outcomes. A positive association was found with physical functioning (β-coefficient = 4.52; *p*-value = 0.01), indicating that as CPM increases, physical functioning improves. Similarly, a positive association was found with limitations due to emotional problems (β-coefficient = 6.57; *p*-value = 0.0487), suggesting that higher CPM is linked to greater improvements in these limitations. Emotional well-being also showed a significant positive association (β-coefficient = 3.51; *p*-value = 0.02), implying that as CPM increases, emotional well-being improves. Additionally, the impact of pain on daily life was positively associated with CPM (β-coefficient = 3.64; *p*-value = 0.04), indicating that higher CPM is linked to better scores, reflecting a reduced impact of pain on daily activities.

Conversely, for participants with a BMI ≥ 31.99, no statistically significant associations were found between CPM and physical functioning (*p*-value = 0.9833), limitations due to emotional problems (*p*-value = 0.0990), emotional well-being (*p*-value = 0.2473), or the impact of pain on daily life (*p*-value = 0.5126), suggesting no observable relationships in these subgroups.

## 4. Discussion

### 4.1. Main Findings

In our study, we examined the role of BMI as an effect modifier in the relationship between CPM and various clinical and quality-of-life outcomes. Notably, BMI influenced the associations between CPM and depression, physical functioning, emotional well-being, and pain. Our first key finding was that in participants with a BMI below the cutoff of 31.99 kg/m^2^ (BMI mean), an increase in CPM was significantly associated with improved depression scores, as reflected by the negative association with the HDRS. Additionally, in the lower BMI group, CPM was positively associated with multiple outcomes on the SF-36 health survey, including physical functioning, emotional well-being, and limitations due to emotional problems. These findings suggest that in individuals with a normal (or moderately elevated) BMI, CPM contributes to better physical and emotional functioning, as well as a reduced impact of pain on daily activities. In contrast, these beneficial effects were absent in participants with a BMI ≥ 31.99 kg/m^2^, highlighting the potential negative impact of higher BMI on the modulation of pain and overall well-being.

### 4.2. BMI as an Effect Modifier in the Association Between Conditioned Pain Modulation and Depression (HDRS)

After stratifying the sample using a BMI cutoff of 31.99 kg/m^2^, we found a significant negative association between CPM and HDRS in participants with a BMI < 31.99 kg/m^2^. This result suggests that within this group, as endogenous pain control improves (higher CPM), depression symptoms decrease. In contrast, no significant association was observed in the BMI ≥ 31.99 kg/m^2^ group, indicating that higher BMI may disrupt the beneficial effects of CPM on depression (Figure 1).

Indeed, obesity has been shown to increase the risk of depression. A meta-analysis involving over 58,000 participants found that individuals with obesity had a significantly higher risk of developing depression over time, with an unadjusted odds ratio (OR) of 1.55 (95% CI: 1.22–1.98; *p* < 0.001) [21]. Moreover, studies have indicated a significant prevalence of co-occurring obesity and pain [22], and additional research demonstrates that a higher body mass index (BMI), particularly in the overweight and obese ranges, alters the relationship between conditioned pain modulation (CPM) and clinical outcomes, such as depression and the overall impact of symptoms [9]. It is well established that obesity is associated with chronic inflammation, a condition that plays a major role in the development of metabolic diseases and various other health complications [23]. The chronic inflammation linked to obesity is particularly significant in the brain, affecting regions involved in visceral interception and the dopaminergic–serotonergic pathways. This inflammation impairs the brain’s ability to regulate the descending networks between the brainstem and spinal cord. Furthermore, obesity-related changes in musculoskeletal structures, such as increased joint loading and altered tendon stiffness, could disrupt mechanosensor function, contributing to altered sensory feedback and pain processing. These factors may exacerbate peripheral and central sensitization, further impairing the effectiveness of endogenous pain modulation. Furthermore, in obese patients, reduced physical activity and diminished motor cortex activation further disrupt cortical control over key regions in the endogenous pain modulation system. This downregulation compromises the body’s ability to modulate pain effectively [11,24].

Our study is consistent with trends in the existing literature, emphasizing the crucial interaction between BMI and CPM in the context of OA and its psychological impacts. These findings align with our previous observations in fibromyalgia patients (Lacerda et al., 2024) [9], where individuals with a lower BMI showed more effective endogenous pain control, reflected by an increase in CPM and an association with reduced symptoms of depression. This finding is particularly significant, as OA patients frequently demonstrate impaired endogenous pain modulation.

### 4.3. BMI as an Effect Modifier in the Association Between Conditioned Pain Modulation and SF-36

Significant positive associations between CPM and multiple SF-36 outcomes were found in participants with a BMI < 31.99 kg/m^2^. Specifically, higher CPM was associated with improved physical functioning, indicating that as endogenous pain control improves, physical functioning increases. Similarly, positive associations were observed with limitations due to emotional problems and emotional well-being, suggesting that higher CPM is linked to greater improvements in these emotional health outcomes. Additionally, the impact of pain on daily life showed a significant positive association with CPM, indicating that as CPM increases, the burden of pain on daily activities decreases. In contrast, for participants with a BMI ≥ 31.99 kg/m^2^, no significant associations were found between CPM and physical functioning, emotional problems, emotional well-being, or the impact of pain on daily life. Consistent with the findings from HDRS, these results suggest that a higher BMI may disrupt the beneficial effects of CPM on both physical and emotional functioning, as well as the ability to manage pain effectively in daily life (Figure 2, Figure 3, Figure 4 and Figure 5).

In line with these findings, it is known that pain and disability from KOA extend beyond the physical realm, affecting social connections, relationships, and emotional well-being, ultimately diminishing patients’ quality of life. The treatment of KOA has traditionally focused on pain relief and functional improvement, but there is a growing recognition of the need for psychosocial support in managing the emotional and social impacts of the disease. Addressing quality of life is essential in understanding the overall well-being, disease progression, and intervention effectiveness for KOA patients. Psychological distress, including emotional health aspects, is common among individuals with KOA, often resulting from the chronic pain and functional limitations imposed by the disease. Furthermore, the pain and disability associated with KOA can lead to social isolation and a reduction in activities, which further exacerbate psychological symptoms [25].

With that said, our findings reinforce the importance of considering BMI in the management of chronic pain conditions like KOA. The positive associations between CPM and multiple SF-36 outcomes in participants with a BMI < 31.99 kg/m^2^ underscore that effective endogenous pain control plays a vital role in various aspects of well-being. For physical functioning, the data suggest that as CPM improves, individuals experience better physical capabilities, which can translate into greater mobility and the ability to maintain independence. Regarding emotional well-being, enhanced CPM appears to contribute to reduced psychological distress, highlighting the strong link between pain control and mental health. For limitations due to emotional problems, improved CPM was associated with fewer restrictions in daily activities caused by emotional issues, suggesting that better pain modulation helps mitigate the impact of psychological symptoms on functioning. Lastly, the impact of pain on daily activities was significantly lower with higher CPM, indicating that effective endogenous pain control enables individuals to maintain a more active and fulfilling daily life despite chronic pain. In contrast, for participants with a BMI ≥ 31.99 kg/m^2^, no significant associations were found between CPM and these outcomes. This suggests that higher BMI may disrupt the beneficial effects of CPM on both physical and emotional functioning, as well as the ability to manage pain effectively in daily life. Addressing BMI-related factors alongside pain management may be crucial for improving quality of life in this population.

### 4.4. Strengths and Limitations

This study presents several strengths, including a well-characterized sample and the use of BMI stratification to examine its role as an effect modifier in the relationship between CPM and clinical outcomes. However, the sample demographics introduce some limitations that may affect the generalizability of the findings. The cohort consists predominantly of female participants, with the majority identifying as White, which may not fully represent the broader population affected by similar conditions. Additionally, there was a predominance of overweight and obese individuals, which may have influenced the observed associations between BMI, pain modulation, and quality-of-life outcomes. While this demographic specificity allows for valuable insights into how BMI affects clinical outcomes, further research is necessary to validate these findings across more diverse populations.

## 5. Conclusions

Our study provides valuable insights into the role of BMI as an effect modifier in the relationship between CPM and various clinical and quality-of-life outcomes. Consistent with our previous findings in fibromyalgia patients (Lacerda et al., 2024) [1], these results suggest that obesity disrupts the beneficial effects of endogenous pain modulation, contributing to poorer physical and emotional functioning. This underscores the importance of considering BMI when designing treatment plans for KOA, where effective pain management is essential. By acknowledging the impact of BMI on both physical and psychological health, healthcare providers can develop more personalized and effective interventions that address not only pain but also broader health concerns. Future research should explore the mechanisms through which BMI influences treatment outcomes and focus on creating strategies that integrate physical and psychosocial support for a more holistic approach to care.

## Figures and Tables

**Figure 1 neurosci-06-00014-f001:**
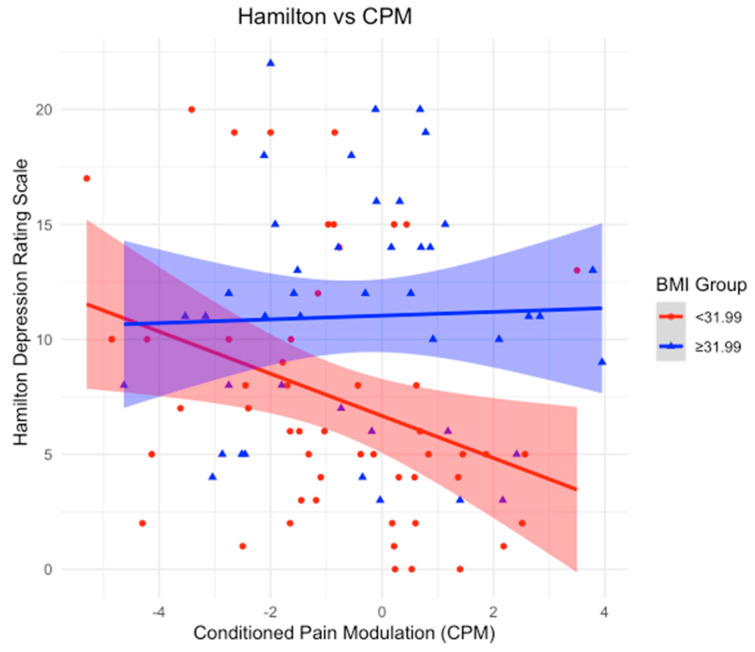
Association between Conditioned Pain Modulation (CPM) and Hamilton Depression Rating Scale (HDRS) scores, stratified by BMI groups. The red line represents individuals with BMI < 31.99, while the blue line represents individuals with BMI ≥ 31.99. Shaded areas indicate 95% confidence intervals. A statistically significant negative association is observed in the lower BMI group, while no clear trend is evident in the higher BMI group, suggesting a potential disruption of CPM’s protective effect on depression in individuals with higher BMI.

**Figure 2 neurosci-06-00014-f002:**
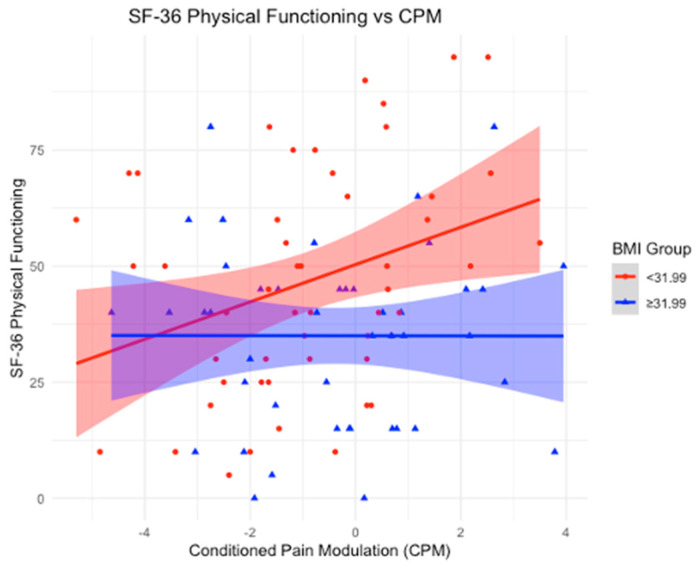
Association between Conditioned Pain Modulation (CPM) and SF-36 Physical Functioning scores, stratified by BMI groups. The red line represents individuals with BMI < 31.99, while the blue line represents individuals with BMI ≥ 31.99. Shaded areas indicate 95% confidence intervals. A statistically significant positive association is observed in the lower BMI group, whereas the higher BMI group shows no clear trend, suggesting a potential disruption in the relationship between CPM and physical functioning in individuals with higher BMI.

**Figure 3 neurosci-06-00014-f003:**
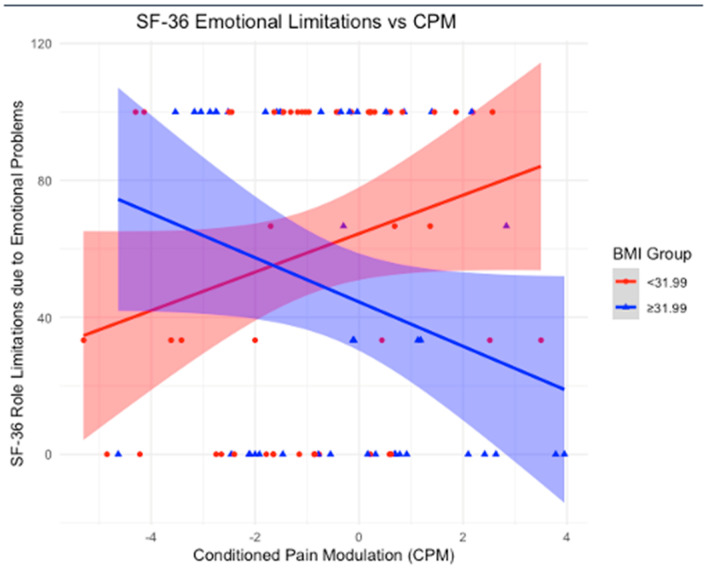
Association between Conditioned Pain Modulation (CPM) and SF-36 Role Limitations due to Emotional Problems scores, stratified by BMI groups. The red line represents individuals with BMI < 31.99, while the blue line represents individuals with BMI ≥ 31.99. Shaded areas indicate 95% confidence intervals. A statistically significant positive association is observed in the lower BMI group, whereas the higher BMI group shows a negative association, suggesting a potential disruption in the relationship between CPM and emotional role functioning in individuals with higher BMI.

**Figure 4 neurosci-06-00014-f004:**
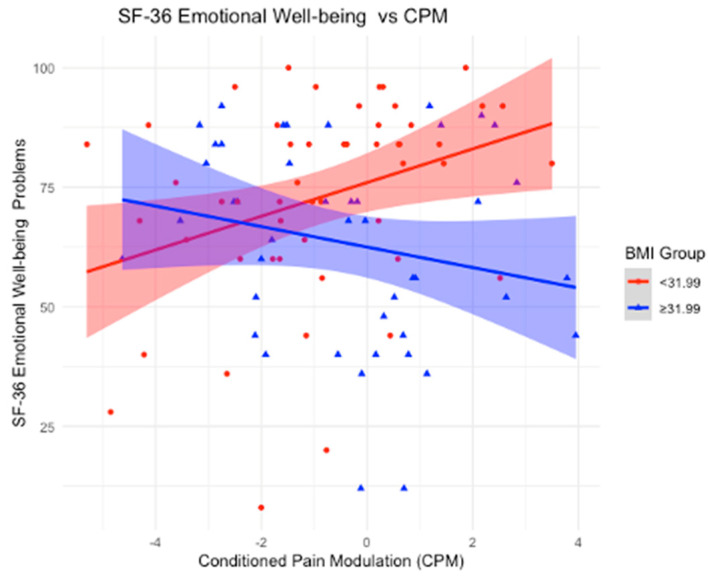
Association between Conditioned Pain Modulation (CPM) and SF-36 Emotional Well-being scores, stratified by BMI groups. The red line represents individuals with BMI < 31.99, while the blue line represents individuals with BMI ≥ 31.99. Shaded areas indicate 95% confidence intervals. A statistically significant positive association is observed in the lower BMI group, whereas the higher BMI group shows a negative association, suggesting a potential disruption in the relationship between CPM and emotional well-being in individuals with higher BMI.

**Figure 5 neurosci-06-00014-f005:**
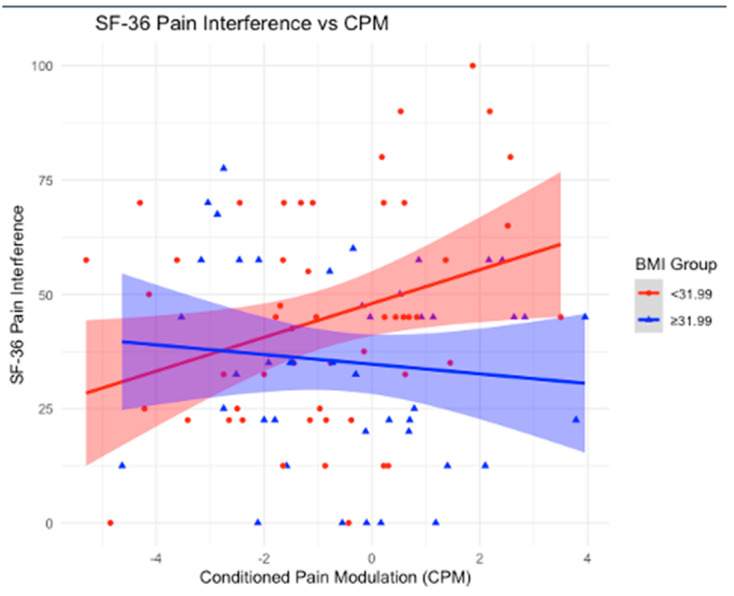
Association between Conditioned Pain Modulation (CPM) and SF-36 Pain Interference scores, stratified by BMI groups. The red line represents individuals with BMI < 31.99, while the blue line represents individuals with BMI ≥ 31.99. Shaded areas indicate 95% confidence intervals. A statistically significant positive association is observed in the lower BMI group, whereas the higher BMI group shows no clear trend, suggesting a potential disruption in the relationship between CPM and pain interference in individuals with higher BMI.

**Table 1 neurosci-06-00014-t001:** Sample demographic characteristics.

Age years: mean, SD	68.7	9.45
Sex: number, %	Males: 19	16.8
	Females: 94	83.2
Education: number, %	Illiterate = 2	1.8
	Elementary school = 48	42.3
	High school = 34	30.1
	College or higher = 29	25.7
Ethnicity: number, %	Asian: 6	5.3
	White: 72	63.7
	Black or African American: 13	11.5
	Multiracial: 22	19.5
Disease duration (months): mean, SD	95.7	98.80
Body mass index (kg/m^2^): mean, SD	31.99	5.30
Body mass index number of subjects	BMI < 31.99 kg/m^2^	57
	BMI > 31.99 kg/m^2^	51
	Missing data	5

**SD**: standard deviation.

**Table 2 neurosci-06-00014-t002:** Variable characteristics.

Variable	Mean (95% CI)	SD
Conditioned Pain Modulation	−0.63 ([−1.03]–[−0.23])	2.01
Hamilton Depression Rating Scale	9.36 (8.31–10.41)	5.58
SF-36—Physical Functioning	40.33 (36.09–44.57)	22.37
SF-36—Role Limitations due to Emotional Problems	51.71 (43.07–60.35)	45.61
SF-36—Emotional Well-being	68.5 (64.57–72.43)	20.74
SF-36—Pain	39.35 (34.95–43.75)	23.23

**CI**: confidence interval; **SD**: standard deviation.

**Table 3 neurosci-06-00014-t003:** Interaction term between BMI and independent variables.

	*p*-Value	β-Coefficient	Adjusted R^2^
Hamilton Depression Rating Scale (HDRS) ^†^			
BMI_mean × CPM	0.07	1.00	0.13
SF36 (SF-36) Physical Functioning ^†^			
BMI_mean × CPM	0.07	−4.02	0.10
SF36 (SF-36) Role Limitations due to Emotional Problems ^†^			
BMI_mean × CPM	0.01	−12.09	0.06
SF36 (SF-36) Emotional Well-being ^†^			
BMI_mean × CPM	0.01	−5.67	0.10
SF36 (SF-36) Pain ^†^			
BMI_mean × CPM	0.04	−4.76	0.07

**BMI_mean**: BMI average; ^†^ dependent variable.

## Data Availability

The data that support the findings of this study are available from the corresponding author upon reasonable request. However, access to the data is restricted due to privacy and confidentiality considerations to protect the identities and sensitive information of the participants involved in the study.

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
