# Peer review of "High Body Mass Index Disrupts the Homeostatic Relationship Between Pain Inhibitory Control and the Symptomatology in Patients with Knee Osteoarthritis—A Cross-Sectional Analysis from the DEFINE Study"

_neurosci, 2025, doi:10.3390/neurosci6010014_

Round 1
Reviewer 1 Report
Comments and Suggestions for Authors
Thank you for the opportunity to review this interesting study, which explores the impact of high BMI on pain in KOS patients.
here are my comments
The introduction of the study would benefit from a more detailed explanation of some key concepts considered in the research, such as conditioned pain modulation
Another point to address is that while the introduction mentions the unclear mechanisms linking obesity to disruptions in pain modulation, the study primarily relies on correlation analysis. This could lead to a mismatch between the study's aim and design, as correlation does not necessarily establish causality.
In the discussion, the authors could further elaborate on the mechanistic aspects of their findings. Additionally, it would be helpful to consider other factors that may contribute to the outcomes observed, such as changes in muscle and tendon function, or more broadly, alterations in musculoskeletal structures in individuals with obesity. These factors could provide valuable insights into the observed differences in pain responses. Or in general mechanosensors alterations at the musculoskeletal level.
Reviewer 2 Report
Comments and Suggestions for Authors
Dear Authors,
I have been given the opportunity to review your paper regarding the relationship between BMI and pain in knee osteoarthritis. It is interesting that a BMI ≥ 31.99 kg/m² can markedly influence the mechanisms of pain perception and, consequently, the therapeutic modulation of pain. Below I report some notes section by section.
TITLE PAGE AND ABSTRACT
- “Lacerda, Guilherme J M 1,2; Fregni, Felipe 1; Battistella, Linamara 2,3; Imamura and Marta 2,3” I think there is something strange with this names list, especially the final part “Imamura and Marta”, so please check it
- the Abstract is fine, but I would avoid the use of citations in it (this means that probably you will have to reallocate citation n.1 and renumber the references)
- also, in the Abstract, please check the text dimension of the part “Instituto de Medicina Fisica e Reabilitacao (IMREA)” which seems slightly larger then the rest
INTRODUCTION
- “KOA typically presents with joint pain, stiffness, and functional impairments, resulting in decreased mobility and diminished quality of life” maybe a recent reference for the clinical description of KOA might be useful
- “As populations continue to age, and obesity rates rise.” This sentence, written in this way, does not make much sense; maybe you meant “As populations continue to age obesity rates tend to concurrently rise.”? (if so, please add a reference for this)
- “Although BMI has been recognized as a potential modifier of clinical outcomes in other conditions, the specific ways in which body weight influences pain modulation in KOA patients are still unclear.” Also here please add some refences about BMI being a modifier clinical outcome for some conditions
- At line 79 you report “(Lacerda et al., 2024)”, but this is unnecessary, since the numbered citation is already there
METHODS
- “To analyze the data, we calculated the sample size based on detecting clinically meaningful differences in pain and CPM, following the methodology from our previous research on fibromyalgia patients [1].” please, when describing the applied methodologies, avoid taking for granted that the person reading this paper has already read your previous works
- At line 125 you report “(Reidler et al., 2012)”; again, you should use the reference system required by MDPI and avoid referring to authors with the “name – year” format
RESULTS
- Generally fine, maybe check the font used for Table 3 (as it differs from the rest of the text)
DISCUSSION
- Generally ok; maybe it is lacking some references to other authors findings, but it is up to you if you want to add some more references
CONSLUSION
- Generally fine, but avoid “(Lacerda et al., 2024)” at line 346 since also here there is already the numbered reference
ACCESSORY SECTIONS
- There is a general “Disclosures” statement that is not following the template of the journal; you should actually provide the mandatory sections “Author Contributions”, “Institutional Review Board Statement”, “Informed Consent Statement”, “Data Availability Statement” and “Conflicts of Interest” (given that the “Acknowledgments” section is not mandatory if there are no people to acknowledge)
REFERENCES
- They are ok but it seems to me they are not following the format required by MDPI; please check it with the editor
In general, the paper is fine and the content are useful and significant. The English language used is quite easy to understand. However there are some errors in following the template provided by the jornal. Moreover, there is a slight lack of references (less than 20) which I think you can easily overcome by following some of my suggestions for the introduction and discussion.
I wish you to be able to publish your paper soon, once the useful corrections will be completed.
Best regards.
